# The effectiveness of diabetes self-management education intervention on glycaemic control and cardiometabolic risk in adults with type 2 diabetes in low- and middle-income countries: A systematic review and meta-analysis

Hasina Akhter Chowdhury[ID][1,2]*, Cheryce L. Harrison[3], Bodrun Naher Siddiquea[1], Sanuki Tissera[1], Afsana Afroz[4], Liaquat Ali[5], Anju E. Joham[3,6], Baki Billah[1]

1 Department of Epidemiology and Preventive Medicine, School of Public Health and Preventive Medicine, Monash University, Melbourne, Australia, 2 Centre for Injury Prevention and Research, Bangladesh (CIPRB), Dhaka, Bangladesh, 3 Monash Centre for Health Research and Implementation–MCHRI, School of Public Health and Preventive Medicine, Monash University, Melbourne, Australia, 4 Department of Biochemistry and Pharmacology, Faculty of Medicine, Dentistry and Health Sciences, The University of Melbourne, Melbourne, Australia, 5 Pothikrit Institute of Health Studies (PIHS), Dhaka, Bangladesh, 6 Departments of Endocrinology and Diabetes, Monash Health, Melbourne, Australia

* hasina.chowdhury@monash.edu

## Abstract

Diabetes mellitus (DM) poses a significant challenge to public health. Effective diabetes self-management education (DSME) interventions may play a pivotal role in the care of people with type 2 diabetes mellitus (T2DM) in low- and middle-income countries (LMICs). A specific up-to-date systematic review is needed to assess the effect of DSME interventions on glycaemic control, cardiometabolic risk, self-management behaviours, and psychosocial well-being among T2DM across LMICs. The MEDLINE, Embase, CINAHL, Global Health, and Cochrane databases were searched on 02 August 2022 and then updated on 10 November 2023 for published randomised controlled trials (RCTs) and quasi-experimental studies. The quality of the studies was assessed, and a random-effect model was used to estimate the pooled effect of diabetes DSME intervention. Heterogeneity ($I^2$) was tested, and subgroup analyses were performed. Egger's regression test and funnel plots were used to examine publication bias. The risk of bias of the included studies was assessed using the Cochrane risk-of-bias tool for randomized trial (RoB 2). The overall assessment of the evidence was evaluated using the Grading of Recommendations Assessment, Development, and Evaluation approach. A total of 5893 articles were retrieved, and 44 studies (n = 11838) from 21 LMICs met the inclusion criteria. Compared with standard care, pooled analysis showed that DSME effectively reduced the HbA1c level by 0.64% (95% CI: 0.45% to 0.83%) and 1.27% (95% CI: -0.63% to 3.17%) for RCTs and quasi-experimental design studies, respectively. Further, the findings showed an improvement in cardiometabolic risk reduction, diabetes self-management behaviours, and psychosocial well-being. This review

**Data Availability Statement:** All relevant data are within the paper and its Supporting Information files.

**Funding:** The author(s) received no specific funding for this work.

**Competing interests:** The authors have declared that no competing interests exist.

suggests that ongoing support alongside individualised face-to-face intervention delivery is favourable for improving overall T2DM management in LMICs, with a special emphasis on countries in the lowest income group.

## Introduction

Diabetes mellitus (DM) is a prevalent public health concern [1], with an estimated 537 million (10.5%) adults aged between 20 to 79 affected globally in 2021 [2]. Among those adults, approximately 90% had type 2 diabetes (T2DM) [2, 3]. T2DM is the primary cause of major micro- and macro-vascular complications contributing to significant adverse clinical sequelae, including premature death [4]. In recent decades, the prevalence of T2DM has escalated more rapidly in low- and middle-income countries (LMICs) compared with high-income countries (HICs), with an estimated 79.4% of the global T2DM population residing in LMICs [2]. In 2021, the estimated global annual cost of diabetes treatment was 966 billion USD [2], imposing a substantial health and economic burden on individuals, their families, and healthcare systems [5–10].

The cornerstone of T2DM management is controlling glycosylated haemoglobin (HbA1c) and optimising cardiometabolic risk factors [11]. Self-management of healthy lifestyle strategies, typically involving optimisation of diet, increasing physical activity, and weight loss in those who are overweight and obese, are recommended as first-line interventions; however, these are highly dependent on individual health literacy, self-efficacy, and motivation [12]. For this reason, diabetes education is crucial in optimising self-management strategies by enhancing knowledge as well as by encouraging and consolidating behaviour-change skills [13, 14]. All of these can be addressed using diabetes self-management education (DSME) intervention [15–17]. DSME intervention includes educating patients through the application of self-care strategies (facilitating with the knowledge, skill and ability) in a cost-effective manner to enhance treatment adherence, diabetes self-management (diabetes knowledge and self-efficacy), lifestyle change (diet, physical activity and weight management where appropriate) and psychological well-being (health-related quality of life [HrQoL]) [15, 18, 19].

Previous systematic reviews and meta-analyses conducted in HICs demonstrate that DSME intervention is associated with improved glycaemic control, diabetes knowledge, self-efficacy, HrQoL [20–22], and reduction in all-cause mortality [23]. This includes a 0.4% reduction in HbA1c, a more than 5 mg/dl reduction in total cholesterol (TC) and a more than 1 mmol/L reduction in fasting blood glucose (FBG) when compared to standard care [24–29]. In addition, DSME intervention in HICs showed positive changes in diabetes-specific knowledge and lifestyle [30]. However, generalising evidence from HICs to LMICs needs to be interpreted with caution given cultural, ethnic, and economic disparities, as well as the variations among study populations [30, 31]. Recent reviews conducted in LMICs demonstrated that DSME intervention, short-term nutrition education and/or lifestyle modification intervention may enhance glycaemic control [30, 32–35] and anthropometric measures [33]. However, to our knowledge, limited attempts have been made in the literature to assess the effectiveness of DSME interventions on a comprehensive outcome measures in LMICs [36–39], which include the effectiveness in the change in diabetes control and cardiometabolic risk, diabetes self-management behaviours and psychosocial well-being. Thus, the aim of the present review is to comprehensively assess the effectiveness of DSME intervention on glycaemic control (eg. HbA1c/FBG), cardiometabolic risk factors (eg. WC, BMI, LDL, HDL, TC, TG, SBP, and DBP),

diabetes self-management behaviours (eg. diabetes knowledge and self-care) and psychosocial well-being (eg. health-related quality of life) among people with T2DM living in LMICs and to explore intervention characteristics, as well as their mode of delivery, frequency, intensity and duration in relation to the improvement in outcomes.

## Methods

This systematic review and meta-analysis was registered with PROSPERO (CRD: 42022364447) and conducted according to the Preferred Reporting Items for Systematic Reviews and Meta-analyses (PRISMA) guidelines [40] (S1 Table).

### Selection criteria

**Inclusion criteria.** The Participant, Intervention, Comparison, Outcome and Study type (PICOS) framework (S2 Table) informed the inclusion and exclusion criteria. Participants included adults with T2DM residing in LMICs. Any form of educational intervention (e.g. self-management intervention with a variety of educational/behavioural components and/or lifestyle modification to diet and exercise) delivered in an LMIC to people with T2DM and targeting diabetes care management compared with standard care/usual care. Outcomes included any one or combination of the following: glycaemic control (HbA1c/fasting blood glucose [FBG]), cardiometabolic risk body mass index (BMI), waist circumference (WC), high-density lipoproteins (HDL), low-density lipoproteins (LDL), triglycerides (TG), total cholesterol (TC), systolic blood pressure (SBP), diastolic blood pressure (DBP), diabetes knowledge, self-efficacy and health-related quality of life (HrQoL). The study types included either RCT or quasi-experimental designs without language or time restrictions.

**Exclusion criteria.** Studies reporting on type 1 diabetes and gestational diabetes were excluded. Qualitative studies, editorials, commentary, reviews and case reports were excluded.

### Search strategy

Five electronic databases (MEDLINE, Embase, CINAHL, Global Health and Cochrane) were searched from their dates of inception through 02 August 2022 and updated on 10 November 2023 (S3 Table) by two authors (HAC and BNS) in consultation with a senior librarian at Monash University. A range of keywords relating to T2DM including educational intervention and model/tools of diabetes care were used, and the list of LMICs was based on the current World Bank Database [41].

### Study selection process

Retrieved articles were stored and managed using the citation software EndNote X20. Following the searches, two authors (HAC and BNS) independently screened all titles as well as abstracts and excluded studies that did not meet the inclusion criteria. A total of 105 articles were selected for a comprehensive full-text review. Following a review for accuracy, two authors (HAC, and BNS) independently reviewed the full text of these 105 articles, and any discrepancy was discussed with a third author (ST) with the supervision of senior author (BB). Finally, a set of 44 articles were selected to determine final article eligibility (Fig 1). A manual search of reference lists of included studies was also performed.

### Study outcomes

The primary outcome of this study was to assess any changes in glycaemic control (i.e. HbA1c or fasting blood glucose [FBG]) after intervention. Secondary outcomes were cardiometabolic

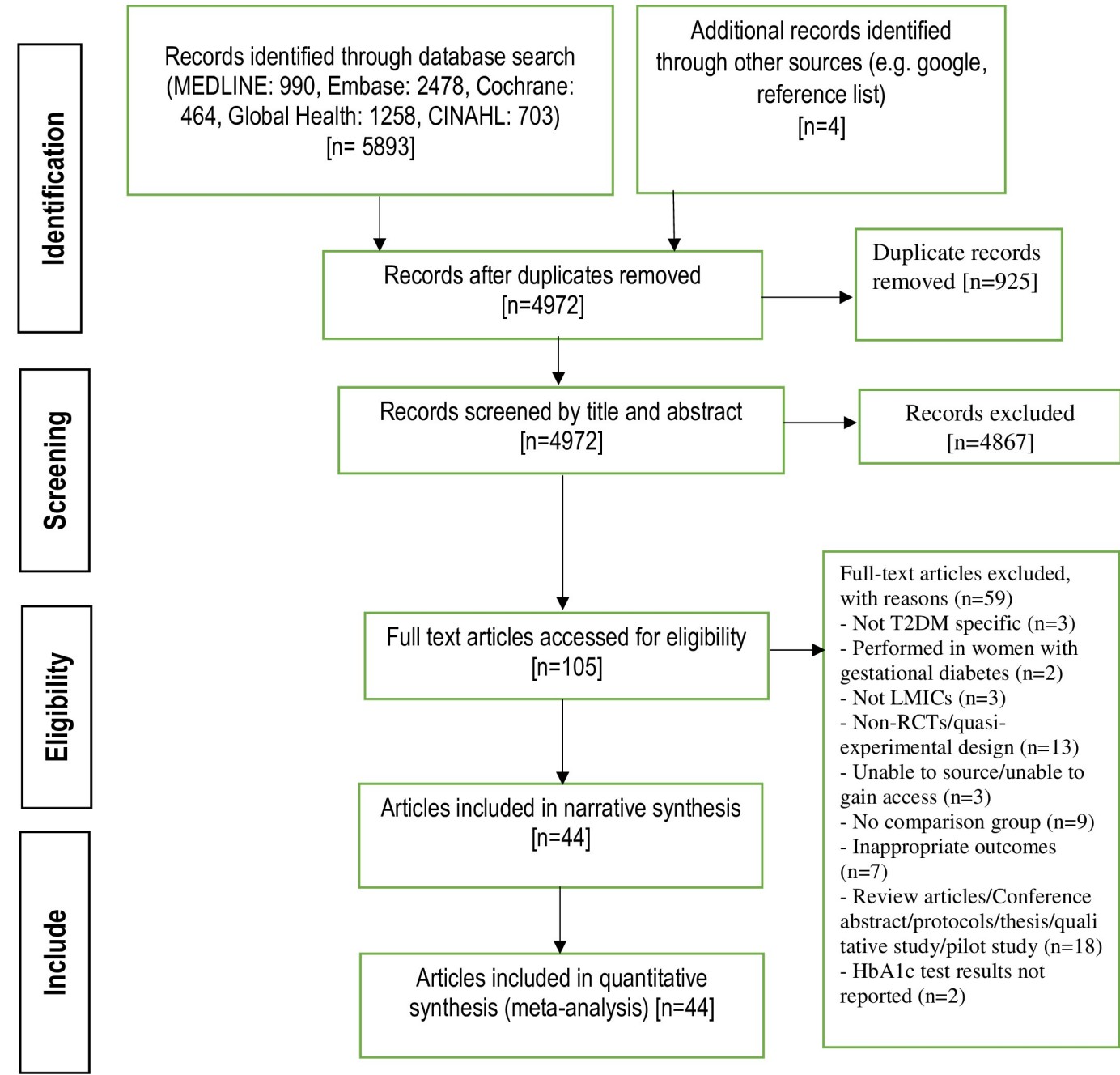

**Fig 1. PRISMA flow diagram.**

risk factors (i.e. BMI, WC, HDL, LDL, TG, TC, SBP or DBP), HrQoL and changes in behavioural outcomes (i.e. diabetes knowledge and self-efficacy [S4 Table]).

### Data extraction

Data from the included articles were extracted independently by two authors (HAC and BNS) using Microsoft Excel. The following information was extracted: publication details (author/s, year of publication and journal), study characteristics (country, study design, setting,

population and sample size), demographics (age of the participants), details of the intervention (type, frequency, intensity, intervention format, duration, number of educational sessions, intervention provider and mode of delivery of the intervention) as well as primary and secondary outcomes (i.e. HbA1c/FBG, BMI, WC, LDL, TG, TC, SBP, DBP, diabetes knowledge, self-efficacy and HrQoL). Discrepancies were discussed and resolved through consensus or arbitration between reviewers.

## Quality assessment

Study quality was appraised independently by two authors (HAC and BNS) using the revised Cochrane risk-of-bias tool for randomised trials (RoB 2) [42, 43] for randomised controlled trials, and the Joanna Briggs Institute (JBI) Critical Appraisal Checklist for quasi-experimental studies (non-randomised experimental studies) [44]. The Cochrane's RoB 2 tool evaluates randomisation process, deviations from the intended interventions, missing outcome data, measurement of the outcome, and selection of the reported result [42]. For this review, the overall risk of bias was rated as high/low/some concerns, in agreement with the RoB 2 tool. Senior author (BB) was consulted to resolve instances of disagreement. A detailed description of the quality assessment has been provided as supporting information (S5 Fig and S6 Table).

## Assessment of certainty of the evidence

Grading of Recommendations, Assessment, Development, and Evaluations (GRADE) was used to evaluate the quality of the evidence [45]. GRADE pro-GDT was employed to summarise the quality of evidence [46]. The certainty of the evidence encompasses consideration of the within-study risk of bias which comprises methodological worth, indirectness of evidence, unexplained heterogeneity, imprecision and, probability of publication bias. The GRADE approach has following four levels of quality such as high-quality evidence that recommends that additional study is very unlikely to change our confidence in the estimate of effect size; moderate quality reflects further research as likely to have a vital impact on the estimate of effect size and may alter the estimate; low quality reveals that further research is very unlikely to have a significant influence on the current estimate of effect size and is likely to change the estimate; and very low quality suggests one is precise indeterminate about the estimate.

## Data analysis

All statistical analyses were performed using Stata V.16 (StataCorp, College Station, Texas, USA). A random-effects model was used to estimate pooled mean differences (MD) for HbA1c or FBG and other relevant quantitative data with a 95% confidence interval (CI). Heterogeneity was tested using the χ2-test on Cochran's Q statistic, which was calculated by means of H and $I^2$ indices. $I^2$ values of over 75% were considered to represent substantial heterogeneity [47]. Subgroup analyses were also performed with the covariates of income level of the country, intervention type, mode of delivery of the intervention and study quality to identify possible sources of heterogeneity. Egger's regression test and funnel plots were used to examine publication bias [48]. As standard deviation of the mean change from baseline is defined as a common missing outcome data [49], and difficulties in running a meta-analysis without missing standard deviations (SDs). The following formula was used to calculate missing SDschange [50]:

$$\text{SDchange} = \sqrt{(\text{SD}^2 \text{ baseline} + \text{SD}^2 \text{ final}) - (2*r*\text{SD baseline}*\text{SD final})}$$. If the SDbaseline and SDfinal values were known, the SDchange value was calculated by assigning a value of 0.7 to the r in the formula, to provide a conservative estimate as undertaken by previous

systematic reviews [50]. All data are reported as a mean difference (95% confidence limits). Characteristics of the included studies are reported as mean (±SD) or number percentages as appropriate. In order to readability of the results, all p-values (where applicable) generated in the tables and forest plots have been approximated to three decimal places while reported in the results section. Statistical tests were considered significant at p-values ≤5% (≤0.05)

## Results

### Selection of studies

A total of 58974 articles were retrieved from the five databases (MEDLINE, Embase, Cochrane, global health and CINAHL) and manual searches. After removing duplicates through title and abstract screening, 105 articles were included for full-text review. Of those, 44 studies (n = 41 RCTs and n = 3 quasi-experimental studies) conducted in 21 LMICs that included 11,838 participants (5,887 in the intervention arm and 5,951 in the comparator arm) (Fig 1).

### Characteristics of the included studies

The characteristics of the included studies are reported in Table 1. Of the 44 studies, 21 were conducted in upper-middle-income countries [51–71], 21 in lower-middle-income countries [1, 38, 72–90], and two were conducted in low-income countries [91, 92], as grouped by the World Bank criteria [41]. The studies were conducted in diabetes clinics or hospitals (n = 15 [34%]), public or private hospitals/clinics (n = 21[48%]) and community settings/home-based locations (n = 8 [18%]). All community settings/home-based studies were conducted in the upper-middle-income countries except one from a low-income country [91]. No community-based studies were conducted in the Southeast Asian region. The HbA1c was reported most frequently (n = 42 [95%] studies), followed by FBG (n = 19 [43%]), BMI (n = 23 [52%]), WC (n = 10 [23%]), LDL (n = 18 [41%]), HDL (n = 17 [39%]), TC (n = 17 [39%]), TG (n = 12 [27%]), SBP (n = 20 [45%]), DBP (n = 17 [39%]), diabetes knowledge (n = 10 [23%]), self-efficacy (n = 7 [16%]), and HrQoL (n = 6 [14%]).

The sample size in the studies ranged from 41 [92] to 1,570 [62], and the average age of the participants was 55 (SD: 6, range 42 to 71 years). The intervention durations ranged from four [59] to 348 weeks [69], with two-thirds (66.6%) of the studies lasting six months in duration. Standard care/usual care comprised the current standard of care as defined by the local programme or setting.

### Intervention characteristics

Overall, the majority of interventions utilised a behaviour-change approach focused on building knowledge, self-efficacy and self-management skills through counselling, coaching, brainstorming or supporting the control of T2DM and its related complications [S5 Table]. Five trials used DM self-management-based coaching programmes [54, 67, 80, 89, 91], four trials used the empowerment approach and interactive teaching model [63, 64, 74, 76], and three used the theory of self-efficacy as a theory or model to underpin intervention content [65, 66, 68]. Each of the following models was used by one trial only: the beliefs, attitudes, subjective norms and enabling factors (BASNEF) model [72]; the predisposing, reinforcing and enabling constructs in educational diagnosis and evaluation (PRECEDE) model [78]; the chronic care model [58]; clinic-based intensified diabetes management model (C-IDM) [60]; the health-belief model [81]; the comprehensive systematic health education and promotion (SHEP) model [85]; the diabetes comprehensive care model (DCCM) [88]; the structured DSME model [38] and the lifestyle intervention holistic model (LIHM) [90]. The remaining 23 trials

**Table 1. Summary characteristics of the included studies.**

| Sl No | First author (year) | Study design | Country | Country by income | Sample size | Study duration (in weeks) | Age in years Mean (SD) | Mode of delivery of the intervention | Intervention format | Model/theory used | Intervention duration; number of sessions (min/session) | Type of intervention | Intervention provider | Settings | Outcome measures |
|---|---|---|---|---|---|---|---|---|---|---|---|---|---|---|---|
| 1 | Askari et al (2018) [72] | Randomised clinical trial | Iran | Lower middle income | 108 (I: 54, C: 54) | 12 | I:66.45 (3.40); C: 67.11 (3.25) | Face to face and telephone follow up | Group session | BASNEF model | 12 weeks; 8 (70) | Lifestyle modification (focus on diet and exercise) | Researcher | Diabetes centre | HbA1c, FBS, TG, LDL, HDL |
| 2 | Azami et al (2018) [93] | Randomised control trial | Malaysia | Upper middle income | 142 (I: 71, C: 71) | 39 | 54.2 (11.8) | Face to face and telephone follow up | Group session | Nurse led DSME (diabetes self-management education) | 12 weeks; 4 (120) | DSME intervention | Nurse | Urban primary and secondary outpatient endocrine clinic within a teaching hospital | HbA1c, TG, HDL, LDL, SBP, DBP, BMI, quality of life, self-efficacy |
| 3 | Baviskar et al (2021), [1] | Randomised control trial | India | Lower middle income | 80 (I: 40, C: 40) | 26 | NR | Face to face | Group session | Self-care and diabetes realted educational intervention | 24 weeks; NR | DSME intervention | Investigator and Medical Social worker | Malavni Urbran Health Training Centre | HbA1c, FBG, BMI, Quality of Life |
| 4 | Chow et al (2016) [51] | Non-clinical randomised controlled trial | Malaysia | Upper middle income | 150 (I:75, C:75) | 26 | NR | Face to face and telephone reminder | Individual session | Home-based educational intervention | 24 weeks; 2 (62) | DSME intervention | Pharmacist | Home based | HbA1c, diabetes knowledge |
| 5 | Debussche et al (2018) [91] | Randomised control trial | Mali | Low income | 151 (I: 76 C: 75) | 52 | I: 53.9 (9.8); C: 51.1 (9.6) | Face to face | Group and individual session | Self-management educational intervention | 52 weeks; 4 (120) | DSME intervention | Peer educators | Community | HbA1c, BMI, SBP, DBP, WC, diabetes Knowledge |
| 6 | Didarloo et al (2016) [73] | Randomised control trial | Iran | Lower middle income | 90 (I: 45, C:45) | 12 | NR | Face to face | Group session | Collaborative and interactive teaching methods | 12 weeks; 4 (60) | DSME intervention | Nurse | Diabetes clinic | HbA1c, Quality of Life |
| 7 | Ebrahimi et al (2016) [74] | Double blind Randomised clinical trial | Iran | Lower middle income | 106 (I:53, C:53) | 8 | I:46.97 (5.54); C:48.15 (6.52) | Face to face | Group session | Empowerment approach training | 8 weeks; 5 to 7 (60 to 90) | DSME intervention | Nurse, endocrinologist and nutritionist | Diabetes center | HbA1c |
| 8 | Essien et al (2017) [75] | Individually-randomised controlled trial | Nigeria | Lower middle income | 158 (I: 59, C:59) | 26 | All: 52.7; I:52.6; C: 52.8 | Face to face and Mobile phone messages | Group session | Diet, nutrition and medication related education | 26 weeks; 12 | DSME intervention | Physician and nurse | Endocrinology clinic, Teaching Hospital | HbA1c |
| 9 | Gathu et al (2018) [76] | Non-blinded randomised clinical trial | Kenya | Lower middle income | 140 (I:70, C:70) | 26 | All: 48.8 (9.8); (I: 50.2 (9.93); C: 47.5 (9.54) | Face to face and telephone reminders | Group session | Diabetes self-management education and support (DSMES): an empowerment and interactive teaching model | 24 weeks; 6 (60) | DSME intervention | Family physician and diabetes educator | Family medicine clinic (private, urban-based) of a university hospital | HbA1c, SBP, DBP, BMI |
| 10 | Goldhaber-Fiebert et al (2003) [52] | Randomised control trial | Vietnam | Upper middle income | 75 (I:40, C:35) | 12 | I: 60 (10); C: 57 (9) | Face to face | Group session | Community-based nutrition and exercise intervention | 12 weeks; 11 (90) | Lifestyle modification (focus on diet and exercise) | Physician | Community centres | HbA1c, FBG, BMI, SBP, DBP, TC, HDLc, LDLc, TG |
| 11 | Goodarzi et al (2012) [77] | Randomised control trial | Iran | Lower middle income | 100 (I:50, C:50) | 12 | I:50.98 (10.32); C: 56.71 (9.77) | Text message | Individual session | Distance education via mobile phone text messaging | 12 weeks; 48 (messages) | DSME intervention | Researcher | Hospital | HBA1c, TC, HDL, LDLc,TG, Knowledge, self-efficacy |
| 12 | Grillo et al (2016) [53] | Single-center, parallel-group, randomised study | Brazil | Upper middle income | 131 (I: 69, C:62) | 54 | I: 61.7 (9.9); C: 63.2 (9.7) | Face to face | Group session | Education on diabetes care | 7 weeks; 7 (120) | DSME intervention | Nurse | Primary care unit | HbA1c, BMI, WC, SBP, DBP, TC, LDL, HDL, TG |
| 13 | Hosseini et al (2017) [78] | Randomised control trial | Iran | Lower middle income | 106 (I:53, C: 53) | 26 | I:51.55 (8.3); C: 58.09 (1.6) | Face to face | Group session | PRECEDE model | 4 weeks; 4 (120) | DSME intervention | General physician and specialist in health education and promotion | Diabetes clinic | HbA1c, BMI |

*(Continued)*

**Table 1.** (Continued)

| Sl No | First author (year) | Study design | Country | Country by income | Sample size | Study duration (in weeks) | Age in years Mean (SD) | Mode of delivery of the intervention | Intervention format | Model/theory used | Intervention duration; number of sessions (min/session) | Type of intervention | Intervention provider | Settings | Outcome measures |
|---|---|---|---|---|---|---|---|---|---|---|---|---|---|---|---|
| 14 | Huo et al (2019) [55] | Randomised clinical trial | China | Upper middle income | 502 (I: 251, C: 251) | 26 | All: 59.5 | Text message | Individual session | A text messaging–based secondary prevention program with the regular automatic delivery of text messages. | 26 weeks; 156 (text messages) | DSME intervention | Text messages | Hospital | HbA1c, FBG, SBP, LDL, BMI |
| 15 | Jain et al (2018) [79] | Open-label randomised controlled trial | India | Lower middle income | 299 (I: 153, C:146) | 24 | I: 55.69 (10.94); C:57.42 (10.95) | Face to face and telephone reminder | Individual session | Combining face-to-face interaction with telephonic reminders by community health workers | 24 weeks; 4 (home visits) | DSME intervention | Community health workers | Tertiary teaching institute | HbA1c, FBS, SBP, DBP, BMI, WC, TC, TG, LDLC, HDL |
| 16 | Jayasuria et al, (2015) [80] | Randomised control trial | Sri Lanka | Lower middle income | 87 (I: 43, C: 42) | 26 | All: 51.4 (7.2) | Face to face | Group and individual session | Diabetes Self-Management-Sri Lanka (DSM-SL) model | 26 weeks; 9 (60) | Lifestyle modification (diet and exercise) | Physician and nurse | Colombo North Teaching Hospital | HbA1c, SBP, TC, LDL, HDL, BMI, self-efficacy |
| 17 | Jiang et al (2019) [56] | Multicentre randomised controlled trial | China | Upper middle income | 265 (I: 133, C: 132) | 26 | All: 56.91 (10.05) | Face to face | Group session | Structured education programme Self-Efficacy for Diabetes (C-SED) Diabetes Distress Scale (C-DDS) Summary of Diabetes Self Care Activities (C-SDSCA) | 26 weeks; 4 (60 to 90) | DSME intervention | Physician and nurse | Multicentre at Bejing, Fujiam, Jiangxi | HbA1c, WC, BMI, blood pressure, TC, TG, LDL, HDL, diabetes knowledge, self-efficacy |
| 18 | Ju et al (2018) [57] | Cluster randomised control trial | China | Upper middle income | 400 (I:200, C:200) | 52 | I: 67.8 (7.4); C: 68.8 (8) | Face to face | Group session | A community based peer support programe | 52 weeks; 12 (120) | DSME intervention | Peer support/ Leaders | Eight community health centres | HBA1c, FBG |
| 19 | Kong et al (2019) [58] | Group Randomized Experimental Study | China | Upper middle income | 278 (I: 142, C: 136) | 39 | I:69.12 (10.54); C: 71.48 (8.79) | Face to face | Group session | Chronic Care Model | 39 weeks; 9 (NR) | DSME intervention | Physician, health manager and public health assistant | Community health service center | HbA1c, SBP, DBP, BMI, TC, LDL, HDL |
| 20 | Lamptey et al (2023) [38] | Single-blind randomised parallel comparator controlled multi-centre trial | Ghana | Lower middle income | 206 (I:103; C:103) | 13 | I: 59; C: 57 | Face to face | Group session | DESMOND: EXTENDing availability of self-management structured education programmes | 13 weeks; 1 (720) | DSME intervention | Educator | Hospitals | HbA1c, WC, SBP, DBP, PAID |
| 21 | Li et al (2016) [59] | Randomized controlled trial | China | Upper middle income | 196 (I: 98, C: 98) | 4 | I:59.1 (4.6); C: 58.3 (4.1) | Face to face | Group session | Structured diet and/ or exercise program (SDEP) | 4 weeks; NR (NR) | DSME intervention | Health educators, doctors, and nutritionists | Hospital | HbA1c, FPG, BMI, TG, TC, HDL, LDL |
| 22 | Lou et al (2020) [60] | Randomised control trial | China | Upper middle income | 1095 (I: 563, C: 532) | 104 | 66.5 (8.7) | Face to face | Group session | Clinic-based intensified diabetes management model (C-IDM) | GPs and nurses: 24 weeks; NR (NR) Patients with diabetes: 78 weeks; 18 (NR) | DSME intervention | Not stated | Disease control centers, general hospitals and local clinics | HbA1c, FBG, SBP, DBP, BMI, TG, TC, HDL, LDL |
| 23 | Mohammadi et al (2018) [81] | A matched-pair design randomized controlled trial | Iran | Lower middle income | 240 (I: 120, C: 120) | 48 | I: 51.2 (6.2); C: 51.4 (6.1) | Face to face | Group session | Health Belief Model (HBM) | 12 weeks; 8 (120) | DSME intervention | Not stated | Golestan Hospital outpatient diabetes clinic | HbA1c, FBS, BMI, TC, TG, LDL, HDL, nutrition knowledge, quality of life, self-efficacy |

(Continued)

**Table 1.** (Continued)

| Sl No | First author (year) | Study design | Country | Country by income | Sample size | Study duration (in weeks) | Age in years Mean (SD) | Mode of delivery of the intervention | Intervention format | Model/theory used | Intervention duration; number of sessions (min/session) | Type of intervention | Intervention provider | Settings | Outcome measures |
|---|---|---|---|---|---|---|---|---|---|---|---|---|---|---|---|
| 24 | Muchiri et al (2016) [61] | Randomised control trial | South Africa | Upper middle income | 82 (I: 41, C: 41) | 52 | I: 59.4 (6.9); C: 58.2 (8.0) | Face to face | Group session | Nutrition education | 52 weeks; 8 (120 to 180) and follow-up 6 (90) | DSME intervention | Health professionals | Community health centres | HbA1c, FBS, BMI, TC, TG, LDL, HDL |
| 25 | Myers et al (2017) [82] | Cluster randomised control trial | India | Lower middle income | 239 (I: 85, C: 154) | 52 | 46.3 (9.5) | Face to face | Group session | Nutrition practice guidelines | 24 weeks; NR (NR) | Lifestyle modification (focus on diet) | Dietitian | Diabetes centres hospitals | HbA1c, BMI, TC, LDL, HDL, TG |
| 26 | Mash et al (2014) [62] | Pragmatic clustered randomized controlled trial | South Africa | Upper middle income | 1570 (I: 710, C: 860) | 52 | I: 55.8 (11.5); C: 56.4 (11.6) | Face to face | Group session | Diabetes education programme | 30 weeks; 4 (60) | DSME intervention | Educator | Community health centres | HbA1c, SBP, DBP, WC, TC, self-efficacy |
| 27 | Ojieabu et al (2017) [83] | Randomised control trial | Nigeria | Lower middle income | 150 (I: 75, C: 75) | 17 | | Face to face | Group session | Intervention of medication and treatment adherence | 17 weeks; 4 (NR) | DSME intervention | Pharmacist | Endocrinology Clinic, Teaching Hospital | FBS, BMI, SBP, DBP |
| 28 | Ramadas et al (2018) [63] | Multi-centre randomised control trial | Malaysia | Upper middle income | 128 (I: 66, C: 62) | 104 | I: 49.6 (10.7); C: 51.5 (10.3) | Web based | Web session | Malaysian Dietary Intervention for People with Type 2 Diabetes: An e-Approach (myDIDeA) | 26 weeks; 12 (12) | Lifestyle modification (focus on diet) | Nutritionist | Public hospital | HbA1c, FBG, diabetes knowledge |
| 29 | Ramli et al (2016) [64] | Pragmatic cluster randomised controlled trial | Malaysia | Upper middle income | 888 (I: 471, C:417) | 104 | I: 58 (0.48); C: 57 (0.5) | Face to face | Group session | EMPOWER-PAR (Participatory action research) interventions | 52 weeks; 2 (NR) | DSME intervention | Physician, nurse, pharmacist and dietitian/nutritionist | Public primary care clinics | HbA1c, BMI, SBP, DBP, WC, TC, TG, LDL, HDL |
| 30 | Samtia et al (2013) [84] | Randomised study | Pakistan | Lower middle income | 344 (I: 174, C: 170) | 20 | I: 46.1; C: 42.3 | Face to face | Group session | Intervention regarding disease knowledge and self-care | 20 weeks; NR (NR) | DSME intervention | Physician and pharmacist | Diabetes clinic at hospital | HbA1c, FBS, BMI |
| 31 | Sanaeinasab et al (2021) [85] | Randomised controlled trial | Iran | Lower middle income | 80 (I: 40, C: 40) | 30 | All: 50.7 (5.9) | Face to face | Group session | Comprehensive systematic health education and promotion (SHEP) model | 7 weeks; 6 (90) | DSME intervention | Not stated | Diabetic clinics | HBA1c, FBG, BMI, SBP, DBP, TC, HDL, LDL, TG |
| 32 | Salahshouri (2018) [86] | Randomised control trial | Iran | Lower middle income | 145 (I: 73; C: 72) | 26 | I: 55.93 (12.4); C: 54.53 (9.43) | Face to face | Group session | Intervention based on psychological factors and nutrition | NR weeks; 8 (60) | Lifestyle modification (focus on diet) | Internal specialists, dietitians, diabetes experts, a psychologist, as well as a religious expert | Diabetic clinics and healthcare centres | HbA1c, FBS, self-efficacy |
| 33 | Tan et al (2011) [65] | Single-blind randomised control trial | Malaysia | Upper middle income | 164 (I:82, C:82) | 12 | I: 54 (9.94); C: 54 (10.74) | Face to face and telephone follow up | Group session | Self-efficacy theory | 12 weeks; 3 (45) | DSME intervention | Not stated | Govt state hospital | HbA1c, diabetes knowledge, self-efficacy |
| 34 | Thanh et al (2021) [87] | Randomised controlled single-center trial | Vietnam | Lower middle income | 364 (I: 182, C: 182) | 52 | All: 62.2 (9.3) | Face to face | Group session | Education on diet, exercise, drug therapy and adherence | 12 weeks; 3 (45) | DSME intervention | Medical staff educators | Diabetes clinic | HbA1c, FBG, SBP |
| 35 | Wattana et al (2007) [66] | Randomised controlled trial | Thailand | Upper middle income | 147 (I:75, C:72) | 26 | I: 58.40 (10.05); C: 55.14 (10.22) | Face to face | Group and individual session | Diabetes self-efficacy and diabetes self-management program | 24 weeks; 5 (90 to 120) and one-off 2 home visits (45) | DSME intervention | Physician and researcher | Diabetic clinics | HbA1c, HrQol |

(*Continued*)

**Table 1.** (Continued)

| Sl No | First author (year) | Study design | Country | Country by income | Sample size | Study duration (in weeks) | Age in years Mean (SD) | Mode of delivery of the intervention | Intervention format | Model/theory used | Intervention duration; number of sessions (min/session) | Type of intervention | Intervention provider | Settings | Outcome measures |
|---|---|---|---|---|---|---|---|---|---|---|---|---|---|---|---|
| 36 | Whittemore et al (2020) [67] | Randomised control trial | Mexico | Upper middle income | 47 (I: 26, C: 21) | 52 | 55.35 (8.75) | Face to face and follow up by phone calls | Group session | Si Yo Puedo DSME program | 52 weeks; 7 (NR) and phone call every 2 weeks and text/picture messages sent daily for 6 months | DSME intervention | Nurse and social worker | Seguro Popular clinics | HbA1c, BMI, SBP, DBP, self-efficacy |
| 37 | Wichit et al (2017) [68] | Randomised controlled trial | Thailand | Upper middle income | 140 (I:70, C:70) | 13 | I: 61.3 (11.6); C: 55.5 (10.5) | Face to face, home visit and telephone follow up | Group session | Self-efficacy theory | 9 weeks; 3 (120) | DSME intervention | Nurse | Hospital | HbA1c, diabetes knowledge, HrQoL |
| 38 | Yan et al (2014) [92] | Randomised study | Mozambique | Low income | 41(I: 31, C: 10) | 12 | I: 53 (2); C: 55 (3) | Face to face | Group session | Exercise training intervention | 12 weeks; 36 to 60 (45) | Lifestyle modification (focus on exercise) | Not stated | Diabetes clinic | HbA1c, BMI, WC, SBP, DBP |
| 39 | Zhang et al (2018) [69] | Randomised study | China | Upper middle income | 998 (I:498, C: 500) | 348 | I: 50.8 (14.3); C: 52.6 (13.2) | Face to face | Group and individual session | Intervention on nutrition therapy, individualized exercise program, screening of complications | 104 weeks; 24 (NR) | DSME intervention | Physician | Hospital | HbA1c, BMI, SBP, DBP, TC, HDL, LDL |
| 40 | Zheng et al (2019) [70] | Randomised controlled trial | China | Upper middle income | 60 (I: 30, C:30) | 104 | 52.22 (11.32) | Face to face | Group session | Diabetes self-management education programme | 104 weeks; 2 (45) | DSME intervention | Therapist guidance | Hospital | HbA1c, FBG |
| 41 | Zhong et al (2015) [71] | Randomised study | China | Upper middle income | 726 (I: 365; C: 361) | 64 | | Face to face | Group session | Peer leader–support program for diabetes management | 24 weeks; 12 (120) | DSME intervention | Peer leaders and staff of Community Health Service Centers (CHSCs) | Community | FBS, BMI, SBP, DBP, diabetes knowledge, self-efficacy |
| 42 | Al-Halaweh et al (2019) [88] | Quasi-experimental study | Palestine | Lower middle income | 200 (I: 100; C: 100) | 52 | I: 56.58 (8.76); C: 57.9 (7.79) | Face to face | Group and individual session | Diabetes comprehensive care model (DCCM) | 52 weeks; 4 (NR) | DSME intervention | Team of internal specialists, dietitians, diabetes experts, psychologist, and religious expert | Mobile diabetes clinic | Wt, Ht, BP, HbA1c, TC, Creatinine, Microalbuminuria |
| 43 | Pamungkas et al (2020) [89] | Quasi-experimental research | Indonesia | Lower middle income | 60 (I: 30; C:30) | 12 | I: 56.5 (7.63); C: 54.2 (9.20) | Face to face | Group session | The diabetes mellitus self-management (DMSM) based coaching program | 12 weeks; 3 (NR) and 1 (home visit) | DSME intervention | Researcher | Public health centers | HbA1c, SBP, DBP, BMI, TC, HDL, LDL |
| 44 | Kumari et al (2018) [90] | Quasi-experimental prospetive trial | India | Lower middle income | 202 (I:102; C: 100) | 65 | I: 51.9 (9.3); C: 54 (8.6) | Face to face | Group and individual session | Lifestyle intervention holistic model (LIHM) | 52 weeks; 6 (10 to 15) | Lifestyle modification (focus on diet) | Dietician, diabetes educator, physical trainer and diabetologist | Delhi Diabetes Research Centre | HbA1c, blood sugar fasting, blood sugar postprandial |

[1, 51–53, 55–57, 59, 61, 62, 69–71, 73, 75, 77, 79, 82–84, 86, 87, 92] cited no theoretical framework or model used to inform the intervention designs.

Approximately 73% (n = 32) of the interventions were delivered using a face-to-face format, 20% (n = 9) utilising face-to-face intervention with telephone follow-up and 7% (n = 3) using a remotely delivered text message/web-based intervention. Intervention was delivered by healthcare professionals (e.g. physician, nurse, pharmacist, health educator, dietitian or nutritionist) in 32 trials [1, 38, 51–54, 56, 58, 59, 61–64, 66–70, 73–76, 78–80, 82–84, 86–88, 90], by the research team in three trials [72, 77, 89], by peer leaders or lay facilitators in three trials [57, 71, 91] and by trained educators in one trial [62]. Five trials did not report the type of intervention facilitator [60, 65, 81, 85, 92]. The intervention formats included groups (n = 33 [75%]), individuals (n = 4 [9%]), a combination of groups and individuals (n = 6 [14%]) and web-based (n = 1 [2%]) intervention strategies.

## Effect of DSME intervention on HbA1c and FBG control

Of 41 RCT studies, 39 reported HbA1c (n = 10,500 participants). Upon meta-analysis, intervention significantly lowered HbA1c levels compared to the control, with a MD of 0.64% (95% CI: 0.64% to 0.83%; p = 0.001). Heterogeneity was very high between the studies (I$^2$ = 94%) with no publication bias (Egger's regression test, p = 0.068) (Fig 2 and Table 2).

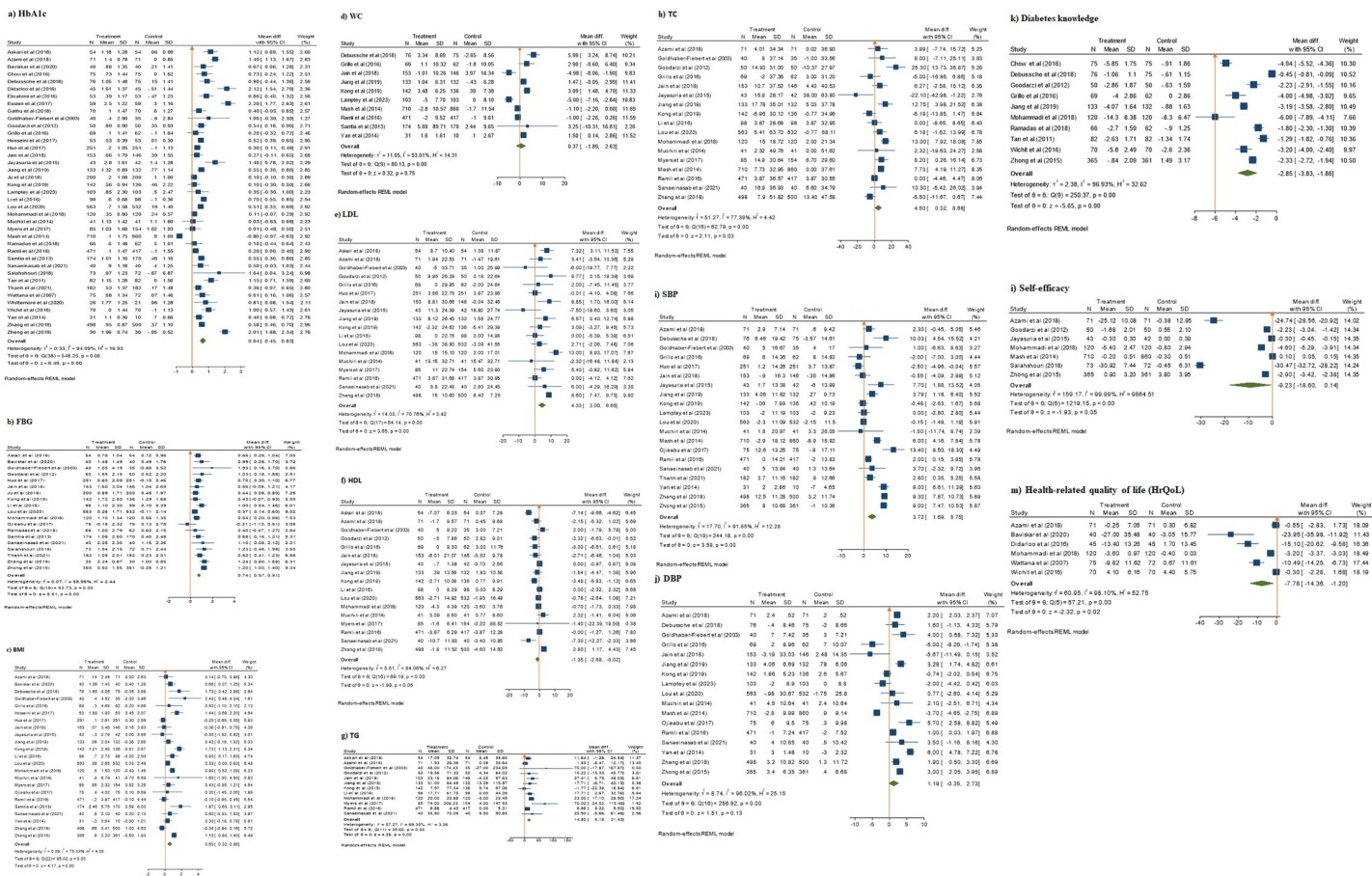

**Fig 2.** Meta-analysis results showing the effect of DSME interventions on clinical outcomes (a) HBA1c (b) FBG (c) BMI (d) WC (e) LDL (f) HDL (g) TG (h) TC (i) SBP (j) DBP, (k) diabetes knowledge, (l) self-efficacy, and (m) health-related quality of life of RCTs studies [Data are reported as mean difference (95% confidence limits)].

**Table 2. Summary results.**

| Study design | Outcome types | Measures | n | Mean change difference (with 95% CI), p-value | Effect of intervention | Heterogeneity ($I^2$ in %) | Publication bias (Egger's regression test p) |
|---|---|---|---|---|---|---|---|
| **RCTs** | Clinical | HbA1c | 39 | 0.64 (0.45, 0.83), 0.001 | Effective | 94 | 0.0680 |
| | | FBG | 19 | 0.74 (0.57, 0.91), 0.001 | Effective | 5996 | 0.5927 |
| | Metabolic risk factors | BMI | 23 | 0.60 (0.32, 0.88), 0.001 | Effective | 75 | 0.1738 |
| | | WC | 10 | 0.37 (-1.89, 2.63), 0.001 | Effective | 93.01 | 0.6884 |
| | | LDL | 18 | 4.33 (2.33–6.65), 0.001 | Effective | 71 | 0.0758 |
| | | HDL* | 17 | -1.35 (-2.69, 0.02), 0.05 | Effective | 84.06 | 0.2715 |
| | | TC | 17 | 4.50 (0.32, 8.68), 0.03 | Effective | 779 | 0.5804 |
| | | TG | 12 | 14.80 (8.18, 21.43), 0.001 | Effective | 69 | 0.0535 |
| | | SBP | 20 | 3.72 (1.69, 5.75), 0.001 | Effective | 92 | 0.8676 |
| | | DBP | 17 | 1.19 (-0.35, 2.73), 0.13 | Effective | 96 | 0.5148 |
| | Diabetes self-managemnt behaviours | Diabetes knowledge* | 10 | -2.85 (-3.83, -1.79), 0.001 | Effective | 97 | 0.0070 |
| | | Self-efficacy* | 7 | -9.23 (-18.60, 0.14), 0.001 | Effective | 99 | 0.0001 |
| | Psychosocial | HrQoL* | 6 | -7.78 (-14.36, -1.20), 0.02 | Effective | 98 | 0.0005 |
| **Quasi-experimental design study** | Clinical | HbA1c | 3 | 1.27 (-0.63, 3.17), 0.19 | Effective | 97 | 0.4515 |

*Negative results consider the positive effect of the intervention

Among 19 studies (n = 5,370 patients) that reported FBG, an overall decrease by 0.74 mmol/L (95% CI: 0.57% to 0.91%; p < 0.001) was observed in the intervention as compared with the control, with moderate heterogeneity ($I^2$ = 59%) and no publication bias (Egger's regression test, p = 0.592) (Table 2).

In trials with quasi-experimental designs, the findings showed a mean reduction in HbA1c of 1.27% (95% CI: -0.63% to 3.17%; p = 0.19) in the intervention as compared to the control (Fig 3). The $I^2$ indicator was 97%, indicating a high heterogeneity with no publication bias (Egger's regression test, p = 0.451) (Table 2). These studies did not report FBG levels.

### Effect of DSME interventions on cardiometabolic risk factors

DSME intervention reduced BMI by 0.60 kg/m$^2$ (95% CI: 0.32% to 0.88%; p = 0.001, $I^2$ = 75.33%) in 23 studies comprising 7,253 participants (Fig 2). Similarly, the results presented in

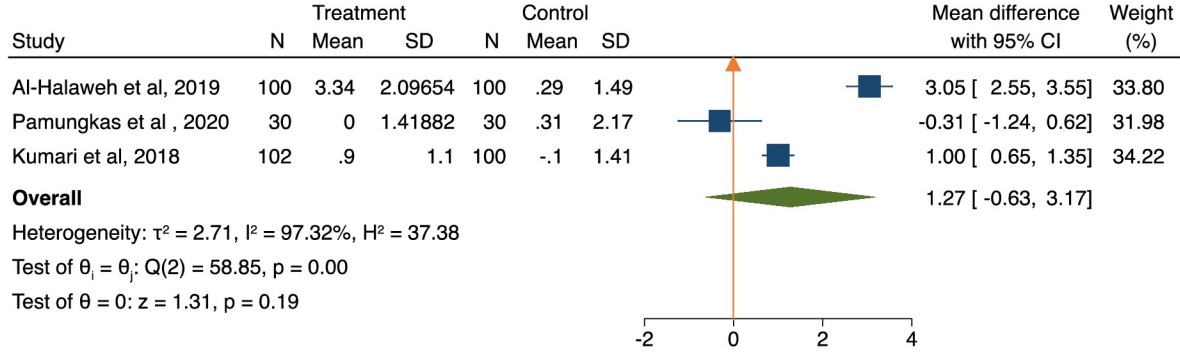

**Fig 3. Meta-analysis results showing the effect of DSME interventions on glycaemic control (HbA1c) of quasi-experimental studies.**

Table 2 and forest plots showed a positive intervention effect on all cardiometabolic risk factors: WC (n = 4,173, MD 0.37, 95% CI: -1.89% to 2.63%; p = 0.001, $I^2$ = 93%), LDL (n = 5803, MD 4.33, 95% CI: 2.33% to 6.65%; p = 0.001, $I^2$ = 71%), HDL (n = 5301, MD -1.35, 95% CI: -2.69% to -0.02%; p = 0.05, $I^2$ = 84.06%), TG (n = 6763, MD 14.80. 95% CI: 8.18% to 21.43%; p < 0.001, $I^2$ = 69%), TC (n = 6,763, MD 4.50, 95% CI: 0.32% to 8.68%; p = 0.03, $I^2$ = 779%), SBP (n = 8,128 MD 3.93, 95% CI: 1.83% to 6.04%; p <0.001, $I^2$ = 926%) and DBP (n = 7,177, MD 1.19, 95% CI: -0.35% to 2.73%; p = 0.13, $I^2$ = 96%). Moderate-to-high heterogeneity was observed across all forest-plot analyses of cardiometabolic risk factors.

## The effect of DSME intervention on diabetes knowledge, self-efficacy and HrQoL

Ten studies (n = 2,195) that evaluated knowledge of diabetes showed an improvement by MD of -2.85 (95% CI: -3.83% to -1.86%; p<0.001, $I^2$ = 97%) with presence of publication bias (Egger's regression test, p = 0. 0.007) (Fig 2). Impact on self-efficacy was addressed in seven studies (n = 1,588), showing an increase by 9.23 (95% CI: -18.60% to 0.14%; p = 0.05, $I^2$ = 99%) with presence of publication bias (Egger's regression test, p = 0.0070) (Fig 2). Six trials (n = 839) that reported HrQoL showed improvement by -7.78 (95% CI: -14.36% to –1.20%; p = 0·02, $I^2$ = 98%). Publication bias was present in these studies (Egger's regression test, p = 0.0005) (Fig 2).

## Subgroup/Sensitivity analysis

Moderate-to-high heterogeneity was observed across the studies regarding primary as well as secondary outcomes. In order to identify the sources of heterogeneity, subgroup/sensitivity analysis was conducted for the DSME intervention by the income level of the country, intervention type, mode of delivery of intervention and quality of the studies. As outlined in S1 Fig, DSME intervention showed that lower-middle-income countries had improvement in HbA1c with a MD of 0.75% (95% CI: 0.45% to 1.06%; p<0.001, $I^2$ = 92%). Further, lifestyle modification (i.e. diet and/or exercise) intervention showed a greater effect on HbA1c reduction (MD: 0.69%, 95% 0.22% to 1.16%; p<0.001, $I^2$ = 78%) than DSME interventions (MD: 0.63%, 95% CI: 0.42 to 0.86; p<0.001, $I^2$ = 95%) (Table 3 and S2 Fig). In addition, subgroup analysis by mode of delivery of intervention showed that face-to-face intervention with periodic telephone follow-up had the highest efficacy on HbA1c reduction (MD: 1.02%, 95% CI: 0.63% to 1.40%; p<0.001, $I^2$ = 86%) followed by face-to-face intervention alone (MD: 0.56%, 95% CI:0.32% to 0.80%; p<0.001, $I^2$ = 95%) and text message or web-based intervention (MD: 0.33%, 95% CI: 0.17% to 0.49%; p = 0.35, $I^2$ = 0.00) (Table 3 and S3 Fig). The quality of the trials with some concerns showed (S4 Fig) reduction in HbA1c with a MD of 0.66% (95% CI: 0.41% to 0.90%, p<0.001, $I^2$ = 93%) compared with trails rated as high or weak. The S1–S4 Figs present subgroup analyses for BMI and lipid profiles (LDL, HDL, TG and TC) by the income level of the country, intervention type, mode of delivery of the intervention and quality of the study. In studies from low-income countries (MD: 0.87, 95% CI: -0.48% to 2.22%; p = 0.05, $I^2$ = 75%), DSME intervention (MD: 0.63, 95% CI: 0.31% to 0.94%; p<0.001, $I^2$ = 78%), face-to-face intervention (MD: 0.71, 95% CI: 0.41% to 1.01%; p<0.001, $I^2$ = 74%) and trials evaluated as high risk (MD: 0.68, 95% CI: 0.18% to 1.18%, p<0.001; $I^2$ = 82%) showed a better BMI reduction. Further, studies conducted in lower-middle income countries presented an improvement in LDL (MD: 7.32%, CI: 3.50% to 11.15%; p = 0.05, $I^2$ = 56%), HDL (MD: -3.12, 95% CI: -5.62% to -0.62%; p<0.001, $I^2$ = 89%), TC (MD:8.72, 95% CI: 0.88% to 18.32%; p<0.001, $I^2$ = 83%) and TG (MD: 21.73, 95% CI: 15.26% to 28.19%; p<0.19, $I^2$ = 10.66%).

**Table 3. Subgroup analysis, based on the income level of the country, intervention type, mode of delivery of the intervention, and quality of the studies.**

| Subgroup | HbA1c | BMI | LDL | HDL | TG | TC |
|---|---|---|---|---|---|---|
| **Income level of the country** | | | | | | |
| Low income | MD: 0.62 (0.13–1.11), $I^2$ 67% | MD: 0.87 (-0.48–2.22), $I^2$ 75% | N/A | N/A | N/A | N/A |
| Lower middle income | MD: 0.75 (0.45–1.06), $I^2$ 92% | MD: 0.69 (0.32–1.06), $I^2$ 46% | MD: 7.32 (3.50–11.15), $I^2$ 56% | MD: -3.12 (-5.62––0.62), $I^2$ 88% | MD: 21.73 (15.26–28.19), $I^2$ 10% | MD: 8.72 (-0.88–18.32), $I^2$ 83% |
| Upper middle income | MD: 0.55 (0.28–0.83), $I^2$ 94% | MD: 0.53 (0.10–0.96), $I^2$ 83% | MD: 2.78 (0.20–6.65), $I^2$ 71% | MD: -0.34 (-1.69–1.00), $I^2$ 69 | MD: 8.85 (8.21–9.48), $I^2$ 0.00% | MD: 2.05 (-1.99–6.09), $I^2$ 660% |
| **Intervention type** | | | | | | |
| Lifestyle modifications (diet and/or exercise) | MD: 0.69 (0.22–1.16), $I^2$ 78% | MD:0.35 (-0.03–0.74), $I^2$ 0.00% | MD:1.63 (-5.58–8.84), $I^2$ 716% | MD: -1.77 (-6.75–3.22), $I^2$ 91% | MD:42.24 (-4.21–88.70), $I^2$ 70% | MD: 0.11 (-17.99–18.22), $I^2$ 78% |
| Self-management | MD: 0.63 (0.42–0.85), $I^2$ 95% | MD:0.63 (0.31–0.94), $I^2$ 78% | MD: 4.33 (2.00–6.65), $I^2$ 71% | MD: -1.14 (-2.38–0.11), $I^2$ 74% | MD:13.64 (6.52–20.77), $I^2$ 69% | MD: 4.86 (0.38–9.35), $I^2$ 77% |
| **Mode of delivery of the intervention** | | | | | | |
| Face-to-face | MD: 0.55 (0.32–0.78), $I^2$ 94% | MD: 0.71 (0.41–1.01), $I^2$ 74% | MD: 3.77 (0.77–6.77), $I^2$ 75% | MD: -0.50 (-1.68–0.68), $I^2$ 76% | MD: 16.93 (8.19–25.68), $I^2$ 74% | MD: 3.15 (-1.08–7.39), $I^2$ 754% |
| Face-to-face and telephone follow up | MD: 1.02 (0.63–1.40), $I^2$ 86% | MD: 0.03 (0.56–0.62), $I^2$ 0.00% | MD: 6.79 (3.58–10.01), $I^2$ 0.00% | MD: -4.18 (-7.46–-0.89), $I^2$ 70% | MD: 11.30 (-1.79–24.39), $I^2$ 62% | MD: 5.44 (-1.62–12.51), $I^2$ 0.00% |
| Text messages or web-based | MD: 0.33 (0.17–0.49), $I^2$ 0.00% | MD: -0.20 (-0.65–0.25), $I^2$ N/A* | MD: 3.87 (-5.51–13.25). $I^2$ 708% | MD: -3.32 (-6.63–0.0.01), $I^2$.%N/A* | MD: 15.22 (-15.33–45.77), $I^2$.% N/A* | MD: 25.30 (13.73–36.87), $I^{2.\%}$ NA* |
| **Quality of the studies** | | | | | | |
| High | MD: 0.60 (0.30–0.91), $I^2$ 94% | MD: 0.68 (0.18–1.18), $I^2$ 82% | MD: 5.40 (-2.26–8.55), $I^2$ 60% | MD: -1.87 (-5.09–1.34), $I^2$ 92% | MD:-2.36 (-10.13–5.42), $I^2$ 71%* | MD: -2.36 (-10.13–5.42), $I^2$ 71% |
| Some concerns | MD: 0.66 (0.41–0.90), $I^2$ 94% | MD: 0.49 (0.19–0.78), $I^2$ 75% | MD: 3.94 (0.79–7.09), $I^2$ 71% | MD: -0.69(-1.31–0.07), $I^2$ 84% | MD: 7.26 (3.00–11.52), $I^2$ 70% | MD: 7.26 (-3.00–11.52), $I^2$ 70% |

*N/A = not applicable, as $\leq$ one study in analysis.

In addition, intervention focused on DSME intervention demonstrated the highest MDs in LDL and TC (LDL: MD 4.33, 95% CI: 2.00% to 6.65%; p<0.001, $I^2$ 71; and TC: MD 4.86 95% CI: 0.38% to 9.35%; p<0.001, $I^2$ 77%) (Table 3). Lifestyle modification intervention alone showed better efficacy in reducing HDL (MD: -1.77, 95% CI: -6.75% to 3.22%; p<0.001, $I^2$ = 91%) and TG (MD 42.24, 95% CI: -4.21 to 88.70; p<0.001, $I^2$ 70%) (Table 3). Furthermore, face-to-face intervention with periodic telephone follow-up showed the highest MDs in LDL (MD 6.79, 95% CI: 3.58% to 10.01%; p = 0.52, $I^2$ = 0.00%) and HDL (MD -4.18, 95% CI: -7.46% to -0.89%; p = 0.03, $I^2$ = 0.03%) (Table 3). However, face-to-face intervention alone was more effective at reducing TG (MD 16.93, 95% CI:8.19% to 25.68%; p<0.001, $I^2$ = 73.96%) (Table 3). Trials classified as high risk of bias showed improvement in the lipid profile of LDL (MD 5.40, 95% CI: -2.26% to 8.55%; p<0.010, $I^2$ = 59.58%), HDL (MD -1.87, 95% CI: -5.09% to 1.34%; p = 0.001, $I^2$ = 92%) and TG (MD 7.26, 95% CI: 3.00% to 11.52%; p = 0.001, $I^2$ = 77%) (Table 3).

## Risk of bias in the included studies

The randomisation process for allocation was evaluated as low risk of bias in 16 studies [1, 30, 52–56, 61, 62, 65, 67, 68, 70, 73, 77, 85], and 13 studies measured as having some concerns of bias [51, 58–60, 63, 64, 75, 79–81, 84, 86, 87]. No trials were rated as low in all five components of the assessment tool. Deviations from the intended interventions were rated as high risk of bias in six studies [57, 69, 72, 82–84]. The risk of bias was rated as some concerns due to missing outcome data in seven studies [51, 59, 71, 76, 77, 85, 93]. Regarding measurement of the outcome reporting, eight studies [54, 69–72, 80, 81, 85, 92] were apparent as high risk of bias.

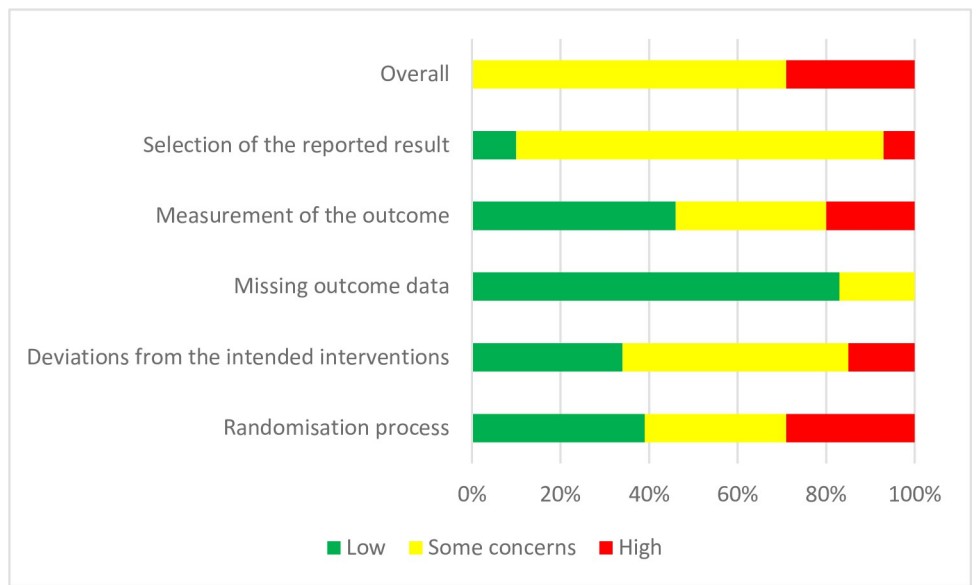

**Fig 4. Risk of bias graph: Review authors' judgements about each risk of bias item presented as percentages across all included studies.**

However, for the selection of the reported results, four studies were evaluated as low risk of bias [53, 74, 86, 91], and three studies were assessed as high risk of bias [58, 75, 93]. The overall risk of bias for studies is summarised in Fig 4 and the risk of bias in individual study is reported in S5 Fig.

A quality assessment was carried out for each of the quasi-experimental studies using the JBI Critical Appraisal Checklist [44, 89, 90]. However, the assessment was a subjective measure that was dependent on the author carrying out the assessment. As per the appraisal checklist, three studies [88–90] were considered and included in the meta-analysis. The details are shown in S6 Table.

## Publication bias

The presence of publication bias for RCTs was visually assessed using a funnel plot for the primary outcome (HbA1c), which showed that there was no publication bias (Table 2). This was supported by the Egger's test (p = 0.0680). Publication bias was also assessed for the secondary outcomes and presented in the Table 2, which showed that there was no publication bias for FBG (p = 0.5927), BMI (p = 0.1738), WC (p = 0. 6884), LDL (p = 0.0758), HDL (p = 0.2715), TC (p = 0.5804), TG (p = 0.0535), SBP (p = 0.8676) and DBP (p = 0.5148). Publication bias, however, was present for HrQoL (p = 0.0005), self-efficacy (p < 0.001) and diabetes knowledge (p = 0.0070). Regarding quasi-experimental studies, no publication bias was observed for HbA1c (p = 0.4515) (Table 2).

## Overall quality of the evidence

The GRADE approach was employed to assess the overall quality of evidence, and the results are summarized in the main comparison's findings. Findings showed that the overall certainty of evidence for HDL and WC were moderate, which suggests further studies will increase our confidence in the estimate of effect size. The quality of the evidence for HbA1c, FBG, and BMI were low, which reflects that the effect size is limited and the true effect may be substantially

different from the estimate of the effect size. The quality of evidence for LDL, TC and TG were very low, which showed that the true effect is probably markedly different from the estimated effect (S7 Table).

## Discussion

This systematic review and meta-analysis aimed to systematically examine the efficacy of DSME interventions on overall T2DM management and cardiometabolic outcomes. Pooled data were used covering 11,838 participants across 44 studies conducted in 21 LMICs. Comprehensive assessment was conducted to evaluate the effectiveness of DSME intervention on 13 outcomes measures including HbA1c control, cardiometabolic risk factors, self-efficacy, diabetes knowledge and psychosocial well-being factors among people with T2DM in LMICs. The outcomes were compared with those generated by standard care across both RCT and quasi-experimental trials. Consequently, a greater number of studies than the earlier reviews were included. This review and meta-analysis demonstrated that DSME intervention leads to better glycaemic control as compared to lifestyle modification intervention alone. Further, it also shows that face-to-face interventions followed by periodic phone calls results in better glycaemic control compared with only face-to-face or remote delivery strategies. The findings suggest that ongoing support is important in optimising intervention efficacy.

Compared with the standard care, this review showed that DSME intervention reduced HbA1c by 0.64% (95% CI: 0.45% to 0.83%) and 1.27% (95% CI: -0.63% to 3.17%) in RCTs and quasi-experimental design studies, respectively. This finding is consistent with previous reviews [20, 21, 93, 94] that reported a reduction in HbA1c levels by 0.83% (95% CI: 1.17% to 0.49%, n = 18 studies) [94] and 0.26% (95% CI: 0.05 to 0.48 n = 31 studies) [25] after DSME interventions. A decrease in HbA1c levels is known to reduce micro- and macro-vascular complications of people with T2DM in long-term follow-up [95–97]. Thus, DSME intervention should be a priority for optimising glycaemic control among people with T2DM in LMICs.

This review demonstrated that DSME intervention leads to significant improvement in FBG and other cardiometabolic risk factors (i.e. BMI, WC, SBP, DBP, LDL, HDL, TG and TC). The findings are in line with those of the previous review that showed the positive effects of group-based self-management education interventions on HbA1c, FBG, body weight, WC, TG and diabetes knowledge [98]. Another review, however, showed that there was no effect of community-based educational interventions on SBP and DBP [99]. Overall, these findings support the potential clinical, behavioural and psychological efficacy of DSME intervention in patients with T2DM.

Adults with diabetes or other metabolic diseases are more likely to have lower self-efficacy, knowledge about their illness and HrQoL [100] as compared with individuals without diabetes and metabolic syndrome. This meta-analysis showed that DSME intervention effectively increased self-efficacy, which is supported by a previous systematic review [101]. Additionally, in a tailored web-based intervention, patients with the highest self-efficacy had better outcomes; therefore, self-efficacy may play a moderating role in intervention outcomes and thus should be considered in tailoring DSME intervention for people with diabetes [102]. Peyrot and Rubin [103] found that those who had the worst self-care, improved the most following DSME intervention and that those with higher self-efficacy had a higher level of self-care behaviours. Self-efficacy provides the confidence necessary to overcoming disease barriers [104] and it receives the most consistent support as a strong determinant of diabetes self-care behaviours [105]. Further, in the present review, diabetes knowledge was significantly improved in the intervention group compared to controls (MD -2.85; 95% CI: -3.83% to -1.79%, p<0.001). Several meta-analyses have similarly shown that DSME interventions are

associated with significant improvements in knowledge of T2DM [94, 106, 107]. Our results also showed that DSME intervention leads to improvement in HrQoL, as reported previously [108]. Other reviews have also demonstrated that DSME and behavioural modification improve HrQoL, which in turn impacts self-care and patients' perceptions about diabetes care [109–112].

Subgroup analyses were performed by the income levels of the countries, intervention types, modes of delivery of the intervention, and quality of the studies. The analysis showed an overall improvement in HbA1c, BMI, LDL, HDL, TG and TC in the LMICs; however, low-income countries had a higher improvement in BMI (MD: 0.87, 95% CI: -0.48 to 2.22). It is possible that health-educational attainment has a direct impact on BMI. In addition, individuals with T2DM in low-income countries may be more physically active due to their need to secure income and also due to limited access to private transportation, leading to a less sedentary lifestyle as compared to those living in lower-middle-income countries [113]. In relation to intervention types, a noteworthy finding in this review is that people with T2DM who received DSME intervention had better BMI, LDL and TC reduction than those who received lifestyle (diet and physical activity) modification alone. This finding is similar to some [33, 34, 114] but not all [10] previous reviews reporting DSME intervention having a better effect on HbA1c control and BMI reduction. In addition to HbA1c and BMI, this current review demonstrated the efficacy of DSME interventions and lifestyle modification intervention in LDL, HDL, TG and TC. Another notable finding of this review is that the face-to-face interventions with periodic telephone follow-up results in better effects on glycaemic control and cardiometabolic risk than face-to-face or text message/web-based interventions alone, which is in line with the National Services Scheme by Diabetes Australia [115]. Periodic phone calls encouraging and reminding patients to practice self-management behaviours consistently over time improves their adherence to overall diabetes control [116]. Thus, face-to-face interventions with periodic telephone follow-up should be prioritised in future DSME intervention programmes for better T2DM management.

This systematic review and meta-analysis is noteworthy in terms of its synthesis of the evidence of outcomes through inclusion of trials using both RCTs and quasi-experimental intervention designs. Overall, it comprehensively summarises the potential clinical, behavioural and psychosocial efficacies of DSME interventions among people with T2DM in LMICs. In addition, five electronic databases were meticulously searched by the authors. As a result, a larger number of trials were identified leading to an impressive sample size of 11,838 participants. This review, however, has a few limitations. First, only a small number of studies were found from low-income countries. Second, the majority of the studies reported outcomes from less than one year follow-up, therefore the long-term effectiveness of DSME intervention in the management of T2DM population cannot be demonstrated. Third, high heterogeneity was observed in the meta-analyses for most of the outcome measures, which is likely due to variation in intervention programme design across the studies [99] as typically noted in intervention programmes of this nature. Fourth, no trial was categorised as low risk in all five components of the ROB 2 assessment tool. Particularly, randomisation process, deviations from the intended interventions, and measurement of the outcome were the most common risks of bias among the RCTs; hence, a prudent approach is warranted when interpreting the results of this present review. It is therefore recommended to follow the CONSORT statement [117] for parallel-group randomised trials to reduce the risk of biases when designing the methodology of the future RCTs. Further, the assessment of outcomes data was measured in heterogeneous ways in the included studies of this review and the certainty of evidence is not sufficient to assert the effectiveness of interventions among patients with T2DM. Hence, to enhance the

certainty of evidence regarding the efficacy of these interventions, future RCTs should address the limitations observed in existing research in the literature.

## Conclusion

In conclusion, this systematic review and meta-analysis may have found a positive effect of DSME on the clinical and cardiometabolic risk factors, diabetes self-management behaviours and psychosocial well-being of people with T2DM in LMICs. Therefore, DSME interventions may enhance disease management and support to improve self-care strategies for people with T2DM. Further, interventions utilising a face-to-face delivery coupled with periodic ongoing support may be useful in improving glycaemic and lipid control as well as anthropometric measures. This study suggests that ongoing support alongside individualised face-to-face intervention delivery needs to be prioritised in order to improve overall T2DM management in LMICs, with a special emphasis on countries in the lowest income groups.

## Supporting information

**S1 Table. PRISMA checklist 2020.**
(DOCX)

**S2 Table. Eligibility criteria (PICOS).**
(DOCX)

**S3 Table. Search strategy.**
(DOCX)

**S4 Table. Primary and secondary outcomes.**
(DOCX)

**S5 Table. Other characteristics (intervention description) of the included studies.**
(DOCX)

**S6 Table. Risk of bias summary for quasi-experimental studies.**
(DOCX)

**S7 Table. GRADEpro level of quality evidence assessment.**
(DOCX)

**S1 Fig.** Subgroup meta-analysis results showing the effect of interventions on (A) HbA1c, (B) BMI, (C) LDL, (D) HDL, (E) TG, and (F) TC based on the income level of the country.
(TIF)

**S2 Fig.** Subgroup meta-analysis results showing the effect of interventions on (A) HbA1c, (B) BMI, (C) LDL, (D) HDL, (E) TG, and (F) TC based on intervention type.
(TIF)

**S3 Fig.** Subgroup meta-analysis results showing the effect of interventions on (A) HbA1c, (B) BMI, (C) LDL, (D) HDL, (E) TG, and (F) TC based on the mode of delivery of intervention.
(TIF)

**S4 Fig.** Subgroup meta-analysis results showing the effect of interventions on (A) HbA1c, (B) BMI, (C) LDL, (D) HDL, (E) TG, and (F) TC based on the quality of study.
(TIF)

**S5 Fig. Risk of bias summary (red, yellow, and green solid circle represents high risk of bias, some concerns risk of bias, and low risk of bias respectively): Review authors**

**judgements about risk of bias item for each included study.**
(TIF)

## Acknowledgments

We would like to acknowledge Mohammad Rocky Khan Chowdhury (PhD Fellow, Department of Epidemiology and Preventive Medicine, Monash University) for technical support.

## Author Contributions

**Conceptualization:** Hasina Akhter Chowdhury, Cheryce L. Harrison, Liaquat Ali, Anju E. Joham, Baki Billah.

**Data curation:** Hasina Akhter Chowdhury, Bodrun Naher Siddiquea, Sanuki Tissera.

**Formal analysis:** Hasina Akhter Chowdhury, Afsana Afroz.

**Investigation:** Hasina Akhter Chowdhury.

**Methodology:** Hasina Akhter Chowdhury, Bodrun Naher Siddiquea.

**Supervision:** Cheryce L. Harrison, Liaquat Ali, Anju E. Joham, Baki Billah.

**Validation:** Cheryce L. Harrison, Baki Billah.

**Writing – original draft:** Hasina Akhter Chowdhury.

**Writing – review & editing:** Hasina Akhter Chowdhury, Cheryce L. Harrison, Bodrun Naher Siddiquea, Sanuki Tissera, Afsana Afroz, Liaquat Ali, Anju E. Joham, Baki Billah.

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
