## [Decision Letter · Decision Letter 0]

26 Sep 2023

PONE-D-23-21626The efficacy of diabetes self-management education intervention on glycaemic control and cardiometabolic risk in adults with type 2 diabetes in low- and middle-income countries: A systematic review and meta-analysis

Dear Dr. Chowdhury,

Thank you for submitting your manuscript to PLOS ONE. After careful consideration, we feel that it has merit but does not fully meet PLOS ONE’s publication criteria as it currently stands. Therefore, we invite you to submit a revised version of the manuscript that addresses the points raised during the review process.

ACADEMIC EDITOR:  please add further detail in the background on the rationale for the review i.e., specifically whether and what similar reviews have been done in HMICs or other LMICs. Be specific as to whether the gap the authors are addressing is the lack of a review focused LMICs in general

Please kindly clearly address the following points :

This systematic review and meta-analysis was registered with PROSPERO (CRD: 42022364447)

a. was the protocol review published ?

b.The PROSPERO registration number did not retrieve the protocol. please check and confirm the PROSPERO registration number

c. Methods: Please provide the comprehensive search strategy for all databases in appendix/ supplementary file.

d.The purpose of the meta-analysis is unclear, particularly in terms of how it relates to the study aim.

e. please add include section of an assessment of the risk of bias of the included studies in this review .I would recommend version 2 of the Cochrane risk-of-bias tool for randomized trials (RoB 2)

Ref: Higgins JPT, Savović J, Page MJ, Elbers RG, Sterne JAC. Chapter 8: Assessing risk of bias in a randomized trial. In: Higgins JPT, Thomas J, Chandler J, Cumpston M, Li T, Page MJ, Welch VA (editors). Cochrane Handbook for Systematic Reviews of Interventions version 6.4 (updated August 2023). Cochrane, 2023. Available from www.training.cochrane.org/handbook.

f. please add the evidence grading section by using the GRADE (Grading of Recommendations, Assessment, Development, and Evaluations)

h.the limitations of the evidence base and the eligible literature are discussed but not the limitations of the review==============================

We look forward to receiving your revised manuscript.

Kind regards,

Mahmoud M Werfalli, PhD

Academic Editor

PLOS ONE

Journal Requirements:

Additional Editor Comments:

please add further detail in the background on the rationale for the review i.e., specifically whether and what similar reviews have been done in HMICs or other LMICs. Be specific as to whether the gap the authors are addressing is the lack of a review focused LMICs in general

This systematic review and meta-analysis was registered with PROSPERO (CRD: 42022364447)

a. was the protocol review published ?

b.The PROSPERO registration number did not retrieve the protocol. please check and confirm the PROSPERO registration number

c. Methods: Please provide the comprehensive search strategy for all databases in appendix/ supplementary file.

d.The purpose of the meta-analysis is unclear, particularly in terms of how it relates to the study aim.

e. please add include section of an assessment of the risk of bias of the included studies in this review .I would recommend version 2 of the Cochrane risk-of-bias tool for randomized trials (RoB 2)

Ref: Higgins JPT, Savović J, Page MJ, Elbers RG, Sterne JAC. Chapter 8: Assessing risk of bias in a randomized trial. In: Higgins JPT, Thomas J, Chandler J, Cumpston M, Li T, Page MJ, Welch VA (editors). Cochrane Handbook for Systematic Reviews of Interventions version 6.4 (updated August 2023). Cochrane, 2023. Available from www.training.cochrane.org/handbook.

f. please add the evidence grading section by using the GRADE (Grading of Recommendations, Assessment, Development, and Evaluations)

h.the limitations of the evidence base and the eligible literature are discussed but not the limitations of the review

Reviewers' comments:

Reviewer's Responses to Questions

**Comments to the Author**

1. Is the manuscript technically sound, and do the data support the conclusions?

Reviewer #1: Partly

Reviewer #2: Yes

2. Has the statistical analysis been performed appropriately and rigorously? 

Reviewer #1: Yes

Reviewer #2: Yes

3. Have the authors made all data underlying the findings in their manuscript fully available?

Reviewer #1: Yes

Reviewer #2: No

4. Is the manuscript presented in an intelligible fashion and written in standard English?

Reviewer #1: Yes

Reviewer #2: Yes

5. Review Comments to the Author

Reviewer #1: I appreciate the effort of the authors. Please find my comments as follows:

1. Methods: Page 4 - Authors have mentioned that they followed the PRISMA guidelines. Authors should follow the updated guidelines (PRISMA 2020). The reference 34 should be updated accordingly. However, the supplementary file shows the PRISMA 2020 checklist. It should be updated in the main text and reference.

2. Methods: Authors have provided the comprehensive search strategy for Medline in appendix. As per the PRISMA 2020 guidelines, the comprehensive search strategy for all databases should be provided in appendix/ supplementary file.

3. In systematic reviews, two independent review authors screen the articles. The authors have mentioned that “Finally, three authors (HAC, BNS and ST) independently reviewed the full text of the remaining records.” This is confusing and needs clarification.

4. In the PRISMA flow diagram, authors have mentioned that “Articles included in qualitative synthesis [n=43]”. This is inappropriate. Qualitative synthesis or qualitative evidence synthesis (QES) denotes a specific method of synthesising qualitative research. Authors should mention “Articles included in synthesis [n=43]” / “Articles included in narrative synthesis [n=43]”.

5. There should be assessment of certainty of evidence using the GRADE (Grading of Recommendations, Assessment, Development, and Evaluations) approach. The 22nd point of PRISMA 2020 checklist is as follows: “Present assessments of certainty (or confidence) in the body of evidence for each outcome assessed.” Authors have mentioned that it has been described in Pages 9, 10, 31. However, in reality, the certainty of evidence has not been assessed. GRADE certainty of evidence should be presented using Summary of Findings (SoF) tables.

6. Abstract: The authors have mentioned that “The risk of bias was evaluated using Eager’s regression test and funnel plot.” This is incorrect. Publication bias was assessed using Eager’s regression test and funnel plot. Risk of bias and publication bias are different.

7. Abstract: Authors should assess the certainty of evidence and provide the certainty level while presenting the results.

8. The search was conducted in August 2022. The search should be updated up to 30 June 2023 (or later).

Reviewer #2: The authors conduct a SR and MA to estimate the effect size of DSME interventions on glycaemic control and CMD risk in LMIC . This is Generally a well written paper and article follows the PRISMA guidelines.

Included below are a few suggestions for improvement.

Title

In the title consider replacing efficacy with effectiveness. DSME interventions tend to be more pragmatic.

Abstract

In the Abstract, consider using reduction in CVD risk rather than improvement in CVD as the later is less subjective

Introduction

Tha investigators site the paucity of data on effectiveness of DSME interventions in LMIC in the introduction. The investigators may find these articles we have recently published useful:

Lamptey R, Amoakoh-Coleman M, Barker MM, Iddi S, Hadjiconstantinou M, Davies M, Darko D, Agyepong I, Acheampong F, Commey M, Yawson A. Change in glycaemic control with structured diabetes self-management education in urban low-resource settings: multicentre randomised trial of effectiveness. BMC Health Services Research. 2023 Dec;23(1):1-9.

Lamptey R, Amoakoh-Coleman M, Djobalar B, Grobbee DE, Adjei GO, Klipstein-Grobusch K. Diabetes self-management education interventions and self-management in low-resource settings; a mixed methods study. Plos one. 2023 Jul 14;18(7):e0286974.

Methods

The methods have been described in sufficient detail to allow reproducibility however the search string has not been provided. If the word limit is a limitation please consider providing the search string and diagrammatic representation of the results of the ROB assessment as part of supplementary materials.. The PROSPERO registration number did not retrieve the protocol. Kindly check and confirm the PROSPERO registration number . Please provide also the date the review was registered.

Results: To improve the readability of the results investigators should consider rounding numbers greater than 10 to whole numbers; p-values can be presented to 2 decimal places at a maximum instead of 4. There are several statements where the authors fail to provide a reference to support the results. The articles included in this systematic review and meta-analysis should be referenced when ever the authors refer to them.

Discussion

Includes relevant literature and situates the findings well. The discussions stem from the results presented and provide adequate interpretation of the findings.

Conclusion

The conclusions are stated too strongly given the limitations of the review e.g the results of the ROB of included studies. Investigators may consider hedging e.g. they study MAY have found a positive effect...

6. PLOS authors have the option to publish the peer review history of their article (what does this mean?). If published, this will include your full peer review and any attached files.

Reviewer #1: **Yes: **KM Saif-Ur-Rahman

Reviewer #2: **Yes: **Roberta Lamptey

---

## [Author Response · Author response to Decision Letter 0]

6 Dec 2023

ACADEMIC EDITOR: 

 Please add further detail in the background on the rationale for the review i.e., specifically whether and what similar reviews have been done in HMICs or other LMICs. Be specific as to whether the gap the authors are addressing is the lack of a review focused LMICs in general

Reply: We have revised the introduction section as per your suggestions.

Please kindly clearly address the following points:

a Comment: This systematic review and meta-analysis was registered with PROSPERO (CRD: 42022364447). was the protocol review published?

Reply: The protocol review has not been published.

b Comment: The PROSPERO registration number did not retrieve the protocol. please check and confirm the PROSPERO registration number.

Reply: The systematic review was registered under the title ‘Impact of an education intervention on type 2 diabetes mellitus in low to middle-income countries: A Systematic Review and Meta-analysis’. However, the revised title was ‘The efficacy of diabetes self-management education intervention on glycaemic control and cardiometabolic risk in adults with type 2 diabetes in low- and middle-income countries: A systematic review and meta-analysis.’ The PROSPERO registration number was CRD42022364447 dated 03 October 2022 and the title was last edited on 01 November 2023.

c Methods: Please provide the comprehensive search strategy for all databases in appendix/ supplementary file.

Reply: We have included all search strategies for five databases in the supplementary file, Table S3 Search strategy.

d Comment: The purpose of the meta-analysis is unclear, particularly in terms of how it relates to the study aim.

Reply: Thank you for your concerns. We appreciate the opportunity to clarify the purpose of the meta-analysis in relation to the study’s aim. The primary objective of our study is to comprehensively assess the effectiveness of diabetes self-management education (DSME) interventions on glycaemic control (eg. HbA1c/FBG), cardiometabolic risk factors (eg. WC, BMI, LDL, HDL, TC, TG, SBP, and DBP), diabetes self-management behaviours (eg. diabetes knowledge and self-care) and psychosocial well-being (eg. health-related quality of life) among people with T2DM living in LMICs. The inclusion of a meta-analysis is integral to achieving this aim as it allows us to synthesis and analyse data from diverse sources, providing a comprehensive overview of the existing literature on DSME education interventions among people with T2DM. Through this meta-analysis, we aim to explore intervention characteristics, as well as their mode of delivery, frequency, intensity, and duration about the improvement in outcomes, and gaps in the literature, eventually contributing to a more robust conclusion. The findings of these this meta-analyses provided a more objective appraisal of the evidence than a narrative review. We have revised the manuscript (last paragraph of the introduction section) to clarify the study objectives and the purpose of the meta-analysis.

e Comment: Please add include section of an assessment of the risk of bias of the included studies in this review. I would recommend version 2 of the Cochrane risk-of-bias tool for randomized trials (RoB 2)

Ref: Higgins JPT, Savović J, Page MJ, Elbers RG, Sterne JAC. Chapter 8: Assessing risk of bias in a randomized trial. In: Higgins JPT, Thomas J, Chandler J, Cumpston M, Li T, Page MJ, Welch VA (editors). Cochrane Handbook for Systematic Reviews of Interventions version 6.4 (updated August 2023). Cochrane, 2023. Available from www.training.cochrane.org/handbook.

Reply: Many thanks for your suggestions. In the revised version, we have included the Cochrane risk-of-bias tool for randomized trials (RoB 2) as recommended. Please see the ‘Quality Assessment’ in the ‘Method’ section in the main manuscript, figure 3, and Supplementary Figure S3.

f Comment: Please add the evidence grading section by using the GRADE (Grading of Recommendations, Assessment, Development, and Evaluations).

Reply: We have added a section for the evidence grading section by using the GRADE (Grading of Recommendations, Assessment, Development, and Evaluations), please see the ‘Assessment of certainty of the evidence’ in the ‘Method’ section in the main manuscript and Supplementary Table S7. 

h Comment: the limitations of the evidence base and the eligible literature are discussed but not the limitations of the review.

Reply: We have addressed the limitations of this review in the main manuscript. Please see the paragraph before the ‘conclusion’ section. 

i Additional Editor Comments: Please add further detail in the background on the rationale for the review i.e., specifically whether and what similar reviews have been done in HMICs or other LMICs. Be specific as to whether the gap the authors are addressing is the lack of a review focused LMICs in general.

Reply: We have revised the introduction section as per your suggestions.

Review Comments to the Author

Reviewer #1: I appreciate the effort of the authors. Please find my comments as follows:

1 Methods: Page 4 - Authors have mentioned that they followed the PRISMA guidelines. Authors should follow the updated guidelines (PRISMA 2020). The reference 34 should be updated accordingly. However, the supplementary file shows the PRISMA 2020 checklist. It should be updated in the main text and reference.

Reply: Thank you for your comment. Reference 34 has been updated accordingly in the main text and reference section.

2 Methods: Authors have provided the comprehensive search strategy for Medline in appendix. As per the PRISMA 2020 guidelines, the comprehensive search strategy for all databases should be provided in appendix/ supplementary file. 

Reply: As per the PRISMA 2020 guidelines, the comprehensive search strategy for all five databases has been provided in the supplementary Table S3. 

3 In systematic reviews, two independent review authors screen the articles. The authors have mentioned that “Finally, three authors (HAC, BNS, and ST) independently reviewed the full text of the remaining records.” This is confusing and needs clarification.

Reply: Revised as comments. The following text has been added to the ‘Study selection process’ in the ‘Method’ section.

‘Following the searches, two authors (HAC and BNS) independently screened all titles as well as abstracts and excluded studies that did not meet the inclusion criteria. A total of 105 articles were selected for a comprehensive full-text review. Two authors (HAC, and BNS) independently reviewed the full text of these 105 articles, and any discrepancy was discussed with a third author (ST) with the supervision of the senior author (BB). Finally, a set of 44 articles were selected to determine final article eligibility’.

4 In the PRISMA flow diagram, authors have mentioned that “Articles included in qualitative synthesis [n=43]c”. This is inappropriate. Qualitative synthesis or qualitative evidence synthesis (QES) denotes a specific method of synthesising qualitative research. Authors should mention “Articles included in synthesis [n=43]” / “Articles included in narrative synthesis [n=43]”.

Reply: We have revised this as per the feedback in the main manuscript.

5 There should be assessment of certainty of evidence using the GRADE (Grading of Recommendations, Assessment, Development, and Evaluations) approach. The 22nd point of PRISMA 2020 checklist is as follows: “Present assessments of certainty (or confidence) in the body of evidence for each outcome assessed.” Authors have mentioned that it has been described in Pages 9, 10, 31. However, in reality, the certainty of evidence has not been assessed. GRADE certainty of evidence should be presented using Summary of Findings (SoF) tables.

Reply: We have included the certainty of evidence and provided the evidence grading section in the main manuscript using the GRADE (Grading of Recommendations, Assessment, Development, and Evaluations). GRADE certainty of evidence has been presented using Summary of Findings (SoF) tables as supplementary materials, see supplementary Table S7.

6 Abstract: The authors have mentioned that “The risk of bias was evaluated using Eager’s regression test and funnel plot.” This is incorrect. Publication bias was assessed using Eager’s regression test and funnel plot. The risk of bias and publication bias are different.

Reply: Thank you for the correction. We have revised the text in the abstract.

7 Abstract: Authors should assess the certainty of evidence and provide the certainty level while presenting the results.

Reply: Thank you for your suggestions. We have included the certainty of evidence and provided the evidence grading section by using the GRADE (Grading of Recommendations, Assessment, Development, and Evaluations) approach in the main manuscript. The following text has been added in the results section:

‘Grading of Recommendations, Assessment, Development, and Evaluations (GRADE) was used to evaluate the quality of the evidence. GRADE pro-GDT was employed to summarise the quality of evidence. The certainty of the evidence encompasses consideration of the within-study risk of bias which comprises methodological worth, indirectness of evidence, unexplained heterogeneity, imprecision, and, probability of publication bias. The GRADE approach has following four levels of quality such as high-quality evidence that recommends that additional study is very unlikely to change our confidence in the estimate of effect size; moderate quality reflects further research as likely to have a vital impact on the estimate of effect size and may alter the estimate; low quality reveals that further research is very unlikely to have a significant influence on the current estimate of effect size and is likely to change the estimate; and very low quality suggests one is precise indeterminate about the estimate’.

8 The search was conducted in August 2022. The search should be updated up to 30 June 2023 (or later).

Reply: We have updated the search till 10 November 2023 and have now included this in the main manuscript.

Reviewer #2: The authors conduct a SR and MA to estimate the effect size of DSME interventions on glycaemic control and CMD risk in LMIC. This is Generally a well written paper and article follows the PRISMA guidelines.

Included below are a few suggestions for improvement.

 Title In the title consider replacing efficacy with effectiveness. DSME interventions tend to be more pragmatic.

Reply: Thank you for the suggestion. We have revised the title in the main text. 

 Abstract

In the Abstract, consider using reduction in CVD risk rather than improvement in CVD as the later is less subjective. 

Reply: We have revised this in the abstract.

 Introduction

Tha investigators site the paucity of data on effectiveness of DSME interventions in LMIC in the introduction. The investigators may find these articles we have recently published useful:

Lamptey R, Amoakoh-Coleman M, Barker MM, Iddi S, Hadjiconstantinou M, Davies M, Darko D, Agyepong I, Acheampong F, Commey M, Yawson A. Change in glycaemic control with structured diabetes self-management education in urban low-resource settings: multicentre randomised trial of effectiveness. BMC Health Services Research. 2023 Dec;23(1):1-9.

Lamptey R, Amoakoh-Coleman M, Djobalar B, Grobbee DE, Adjei GO, Klipstein-Grobusch K. Diabetes self-management education interventions and self-management in low-resource settings; a mixed methods study. Plos one. 2023 Jul 14;18(7):e0286974.

Reply: Many thanks for sharing the articles. They were very helpful. We have included them as references.

 Methods

The methods have been described in sufficient detail to allow reproducibility however the search string has not been provided. If the word limit is a limitation please consider providing the search string and diagrammatic representation of the results of the ROB assessment as part of supplementary materials. The PROSPERO registration number did not retrieve the protocol. Kindly check and confirm the PROSPERO registration number. Please provide also the date the review was registered.

Reply: We have now included all search strategies as supplementary materials (Table S3). The systematic review was registered under the title ‘Impact of an education intervention on type 2 diabetes mellitus in low to middle-income countries: A Systematic Review and Meta-analysis’. However, the revised title was ‘The efficacy of diabetes self-management education intervention on glycaemic control and cardiometabolic risk in adults with type 2 diabetes in low- and middle-income countries: A systematic review and meta-analysis.’ The PROSPERO registration number was CRD42022364447 dated 03 October 2022 and last edited on 01 November 2023.

In addition, we have used the Cochrane risk-of-bias tool for randomized trials (RoB 2) as recommended.

 Results: To improve the readability of the results investigators should consider rounding numbers greater than 10 to whole numbers; p-values can be presented to 2 decimal places at a maximum instead of 4. There are several statements where the authors fail to provide a reference to support the results. The articles included in this systematic review and meta-analysis should be referenced whenever the authors refer to them.

Reply: Many thanks for your suggestions. We have revised these as per feedback in the results section. In order to readability of the results, all p-values (where applicable) generated in the tables and forest plots have been approximated to two decimal places while reported in the results section. In addition, we included all references where applicable. 

 Discussion

Includes relevant literature and situates the findings well. The discussions stem from the results presented and provide adequate interpretation of the findings.

Reply: Thank you for your feedback. 

 Conclusion

The conclusions are stated too strongly given the limitations of the review e.g the results of the ROB of included studies. Investigators may consider hedging e.g. they study MAY have found a positive effect...

Reply: Thank you for your concerns. We have revised the conclusion section as per the suggestions.

---

## [Decision Letter · Decision Letter 1]

4 Jan 2024

The effectiveness of diabetes self-management education intervention on glycaemic control and cardiometabolic risk in adults with type 2 diabetes in low- and middle-income countries: A systematic review and meta-analysis

PONE-D-23-21626R1

Dear Dr. Hasina 

We’re pleased to inform you that your manuscript has been judged scientifically suitable for publication and will be formally accepted for publication once it meets all outstanding technical requirements.

Kind regards,

Mahmoud M Werfalli, PhD

Academic Editor

PLOS ONE

Reviewer #2: All comments have been addressed

2. Is the manuscript technically sound, and do the data support the conclusions?

Reviewer #2: Yes

3. Has the statistical analysis been performed appropriately and rigorously? 

Reviewer #2: Yes

4. Have the authors made all data underlying the findings in their manuscript fully available?

Reviewer #2: Yes

5. Is the manuscript presented in an intelligible fashion and written in standard English?

Reviewer #2: Yes

6. Review Comments to the Author

Reviewer #2: No further comments. All comments raised have been sufficiently addressed. The paper is much stronger

7. PLOS authors have the option to publish the peer review history of their article (what does this mean?). If published, this will include your full peer review and any attached files.

Reviewer #2: **Yes: **Dr Roberta Lamptey

---

## [Editor Report · Acceptance letter]

14 Jan 2024

PONE-D-23-21626R1 

PLOS ONE

Dear Dr. Chowdhury, 

I'm pleased to inform you that your manuscript has been deemed suitable for publication in PLOS ONE. Congratulations! Your manuscript is now being handed over to our production team.

Kind regards, 

on behalf of

Dr. Mahmoud M Werfalli 

Academic Editor

PLOS ONE